# Asymmetric conformational maturation of HIV-1 reverse transcriptase

Xunhai Zheng, Lalith Perera, Geoffrey A Mueller, Eugene F DeRose, Robert E London*

Genome Integrity and Structural Biology Laboratory, National Institute of Environmental Health Sciences, National Institutes of Health, Research Triangle Park, United States

**Abstract** HIV-1 reverse transcriptase utilizes a metamorphic polymerase domain that is able to adopt two alternate structures that fulfill catalytic and structural roles, thereby minimizing its coding requirements. This ambiguity introduces folding challenges that are met by a complex maturation process. We have investigated this conformational maturation using NMR studies of methyl-labeled RT for the slower processes in combination with molecular dynamics simulations for rapid processes. Starting from an inactive conformation, the p66 precursor undergoes a unimolecular isomerization to a structure similar to its active form, exposing a large hydrophobic surface that facilitates initial homodimer formation. The resulting p66/p66' homodimer exists as a conformational heterodimer, after which a series of conformational adjustments on different time scales can be observed. Formation of the inter-subunit RH:thumb' interface occurs at an early stage, while maturation of the connection' and unfolding of the RH' domains are linked and occur on a much slower time scale.

## Introduction

HIV reverse transcriptase (RT) plays a multifunctional role in the transformation of viral RNA into dsDNA and represents a primary target for treatment of AIDS. Currently, all of the drugs in clinical use target the mature RT p66/p51 heterodimer, however, a single p66 peptide chain functions as the precursor for each subunit of the RT heterodimer, requiring a complex maturation process that includes subunit-selective elimination of a single ribonuclease H (RH) domain. The need for such a process is a consequence of a metamorphic polymerase domain that is able to adopt different structures in each RT subunit, allowing it to fulfill two different functional roles. The metamorphic polymerase domain reduces the need for additional coding sequences in the HIV gene, consistent with evolutionary pressures on the size of the RNA viral genome (*Belshaw et al., 2007*), while requiring a more complex structural maturation process. Hypotheses for the formation and maturation of the RT homodimer include proposals in which RH domain proteolysis precedes heterodimer formation (*Srivastava et al., 2006*), models in which p66 forms an initially symmetric homodimer followed by RH domain unfolding leading to an asymmetric homodimer (*Anderson and Coleman, 1992*; *Abram and Parniak, 2005*; *Sharaf et al., 2014*), and models in which an initially formed asymmetric homodimer leads to partial RH domain unfolding (*Hostomska et al., 1991*). Until recently, no detailed structural data were available for the p66 monomer and very little structural evidence was available to support or refute any of the above models. Not only does this represent a significant gap in understanding the behavior of an important viral enzyme but also the intermediates involved in heterodimer formation provide potentially useful targets for the development of new interventional strategies.

We recently determined a crystal structure for an isolated p51 monomer mutant and obtained NMR data indicating that the p66 monomer adopts a structure similar to the p51 monomer with an additional RH domain linked by flexible residues unraveled from the connection subdomain

*For correspondence: london@niehs.nih.gov

Competing interests: The authors declare that no competing interests exist.

**eLife digest** Proteins are made up of long chains of building blocks called amino acids. These chains can twist and fold in numerous ways to adopt the specific three-dimensional shapes that enable each protein to perform its role. In recent years, researchers have identified several proteins that can adopt different shapes from the same sequence of amino acids. These are known as metamorphic proteins and each shape may carry out a different role.

HIV is a virus that causes AIDS, an illness that leads to progressive failure of a person's immune system. The virus uses an enzyme called "reverse transcriptase" to copy its genetic material. The enzyme consists of two metamorphic protein subunits that are both derived from the same precursor protein called "p66". One p66 subunit adopts an extended shape that enables it to carry out enzymatic activities. The second is processed into a smaller p51 subunit that is inactive but provides structural integrity to the enzyme.

Zheng et al. have now used nuclear magnetic resonance and other state-of-the-art techniques to analyze the different stages of the conversion of the p66 protein into the mature reverse transcriptase enzyme. The analysis revealed the shape of a single p66 protein molecule, and showed that occasional changes in shape allow one p66 molecule to bind to a second. This means that an immature version of reverse transcriptase contains two p66 subunits with different shapes. The shapes of each of the two subunits then undergo further changes with time. In one of the subunits, competing interactions lead to a molecular tug-of-war that prevents part of the protein from adopting its folded shape. This part subsequently unravels and is later destroyed by another HIV enzyme (called HIV protease) to form the smaller p51 subunit.

Since HIV needs reverse transcriptase in order to multiply and cause infection, drugs that prevent this enzyme from working are used to treat patients with AIDS. Current drugs target the mature form of the enzyme, but are of limited use because mutations can lead to drug-resistant forms of the proteins. The findings of Zheng et al. now fill a major gap in our understanding of the intermediate steps that lead to the formation of mature reverse transcriptase. These findings are expected to guide future work aimed at developing new drugs that interfere with maturation instead of blocking activity of the mature enzyme.

C-terminus (*Zheng et al., 2014*). The p66 monomer is the substrate for dimerization, and thus, provides the starting point for analysis of p66/p66' dimer formation and subsequent conformational changes. Structural comparisons of the RT subunits and the p51 monomer indicate that the most significant conformational variations are observed for the palm thumb connecting segment (residues 212–240) and for the connection domain. However, information for the connection domain has been particularly limited by the fact that it has not been possible to study it in isolation.

The series of studies described here was designed to more fully characterize the transformation from monomer to mature heterodimer. Mutagenesis-based assignments of the isoleucine δ-methyl resonances arising from the connection domains provide a more complete description of the changes taking place in this highly plastic region of the protein. This information also provides insight into the coordinated changes that link conformational maturation of the p66' connection' domain to RH' unfolding. We also report molecular dynamics simulations for some of the early isomerization events not directly accessible to our NMR measurements. Using our recently introduced isomerization-restricted p66 mutant, we also demonstrate subunit-selective labeling, which allows us to the study the conformational maturation of the p66' subunit of RT without additional resonances from the p66 subunit, greatly reducing the resonance overlap problem. Although the selective labeling/NMR detection strategies utilized cannot provide an atomic-level description of the entire conformational maturation process, they provide localized snapshots of the environment of the labeled residues that allow us to evaluate specific models for this process, much as crystal structures provide snapshots corresponding to different stages of an enzyme-catalyzed transformation. These studies provide a more complete description of the complex conformational maturation processes leading to formation of the p66/p51 RT heterodimer.

## Results and discussion

### Nomenclature

The complexity and degeneracy of the system requires particular attention to the nomenclature required to distinguish between the sequentially identical subunits. The subunit and associated domains that become committed to developing into p51 and the supernumerary RH domain are indicated by primes, for example, p66', thumb', RH', etc. In some instances, we have used the conformation-dependent labeling introduced previously (*Zheng et al., 2010*) in which we designate p66$\underline{M}$ as the monomer conformation; p66$\underline{E}$ corresponds to the more extended p66 conformation observed in the RT heterodimer; p66$\underline{C}$ corresponds to the p66 subunit that contains the compact and inactively folded polymerase domain (p51$C$) linked to a separate RH domain. Individual resonances can then be identified as $M$, $E$, or $C$ indicating the conformational species to which they correspond. Since the conformation and the associated resonances evolve with time, in a few cases, it was necessary to utilize $C_i$ or $E_i$ for the initially observed resonances associated with the $E$ or $C$ conformations.

RT has two functional domains, polymerase and RNase H, with the polymerase made up of fingers, palm, thumb, and connection subdomains. In order to simplify the presentation, the rigorous distinction of domain vs subdomain has been ignored.

### The conformational selection model

The basic features of the conformational selection model deduced on the basis of earlier NMR, structural, and kinetic studies (*Venezia et al., 2009*; *Braz et al., 2010*; *Zheng et al., 2010*, *2014*) can be described by the relations given below:

$$p66M \xleftarrow{\text{domain rearrangements}} p66E_i \qquad [1a]$$

$$p66M + p66E_i \xleftarrow{K_D} p66E_i/p66C_i \qquad [1b]$$

$$p66E_i/p66C_i \xleftrightarrow{\phantom{xx}} \xleftrightarrow{\phantom{xx}} \xrightarrow{\text{RH' unfolding}} p66E/p66C \qquad [1c]$$

In the above, p66$M$ corresponds to the p66 monomer conformation, p66$Ei$ refers to an initially isomerized structure or ensemble of structures similar to, but not exactly identical with the p66 subunit of RT. The structure of p66$C_i$ is very similar to that of the monomer p66$M$, probably including only small adjustments, for example, in the β7-β8 loop to facilitate interface formation (*Mulky et al., 2007*). There are subsequently a number of conformational adjustments within the dimer, culminating with irreversible RH' unfolding, that complete the conformational maturation process to produce the mature p66$E$/p66$C$ homodimer. The p66$E$/p66$C$ structure is equivalent to an RT heterodimer structure in which all residues on the p66$C$ subunit after ~430 are disordered, exposing the major proteolysis site as well as additional sites susceptible to HIV-1 PR cleavage.

The first two steps of the above process are illustrated schematically in *Figure 1*. A key structural feature of the monomer, represented in the upper left hand corner, is the absence of most interface contacts; only the interface between the discontinuous fingers/palm and the connection remains. Thus, the necessary domain rearrangements required for conformational isomerization are more easily accomplished than would be the case if the process began from either the $E$ or $C$ conformational states. The unimolecular isomerization of the p66 monomer depicted in *Figure 1* requires only the occasional dissociation of the fingers/palm:connection interface. Another important feature of the initial homodimer is that the inter-subunit RH:thumb' interface is not present. The absence of this interface provides ample room for accommodation of the supernumerary RH' domain that is present in the initial homodimer. A third feature of the process represented in *Figure 1* is that the detailed interactions between the two connection domains that are present in the mature RT heterodimer are not yet fully realized in the initial dimer structure. Rather, we suggest that the initial structure is more dependent on non-specific hydrophobic stabilization involving residues on the two connection domains. Since many of the early conformational transitions corresponding to the first two equilibria in *Equation 1* are not directly accessible to the NMR methods used in the present study, we utilized

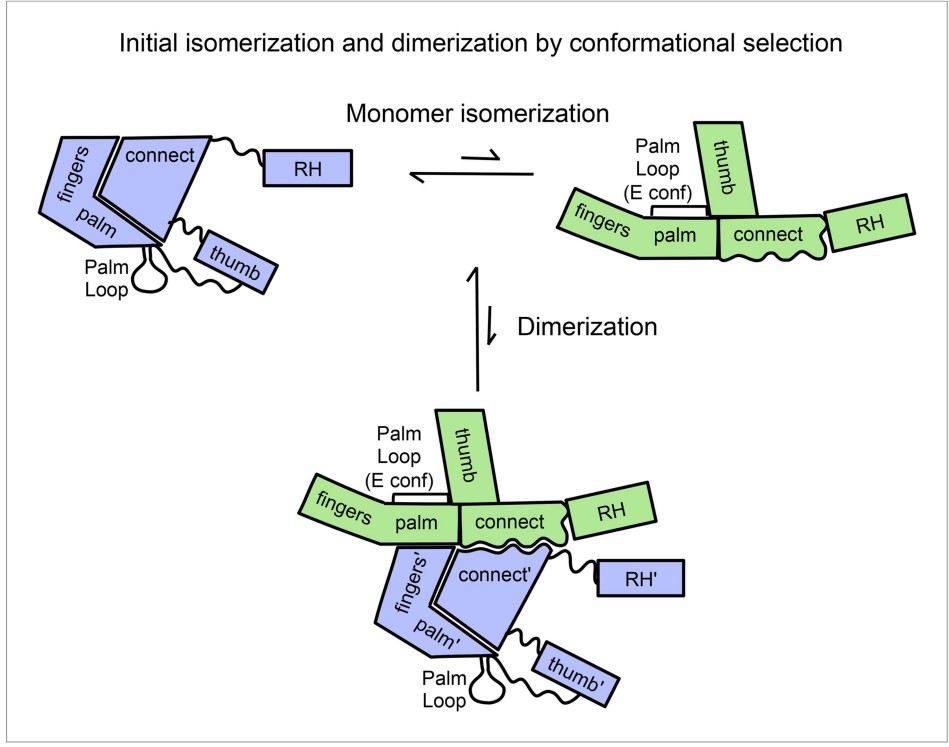

**Figure 1**. Schematic diagram showing proposed isomerization and initial p66 homodimer formation. The subunit conformations are color coded (extended, green; compact, blue). Primes are introduced after homodimer formation to allow subunit identification and indicate the subunit destined to be proteolyzed. The palm loop E conformation becomes the primer grip.

The following figure supplements are available for figure 1:

**Figure supplement 1**. Ribbon diagram representations of reverse transcriptase (RT) monomer and dimer structures.

**Figure supplement 2**. Structural comparison of connection domains.

**Figure supplement 3**. Alternate conformations of helix αM′.

our palm loop deletion mutant as well as molecular dynamics simulations to further probe these initial events.

## Testing the relationship between isomerization and dimerization

The central question related to the initial dimerization event is whether the major reorganization of the polymerase subdomains that interconverts the two subunits of the RT homodimer occurs prior or subsequent to dimer formation. A prior reorganization leads to a conformational selection model (*Equation 1*), while a subsequent reorganization implies an induced fit process that can be described by a version of *Equation 2* below:

$$2p66M \xleftarrow{K_D} p66M/p66M \qquad [2a]$$

$$p66M/p66M \xrightarrow{\text{conformational maturation}} p66E/p66C \qquad [2b]$$

In order to differentiate between these two models, we utilized a p66 deletion mutant, p66ΔPL, lacking palm loop residues 219–230. The residues deleted in this construct are usually disordered in the C conformation of the polymerase domain but play important structural and functional roles forming the primer grip in the E conformation. Thus, this deletion does not significantly interfere with

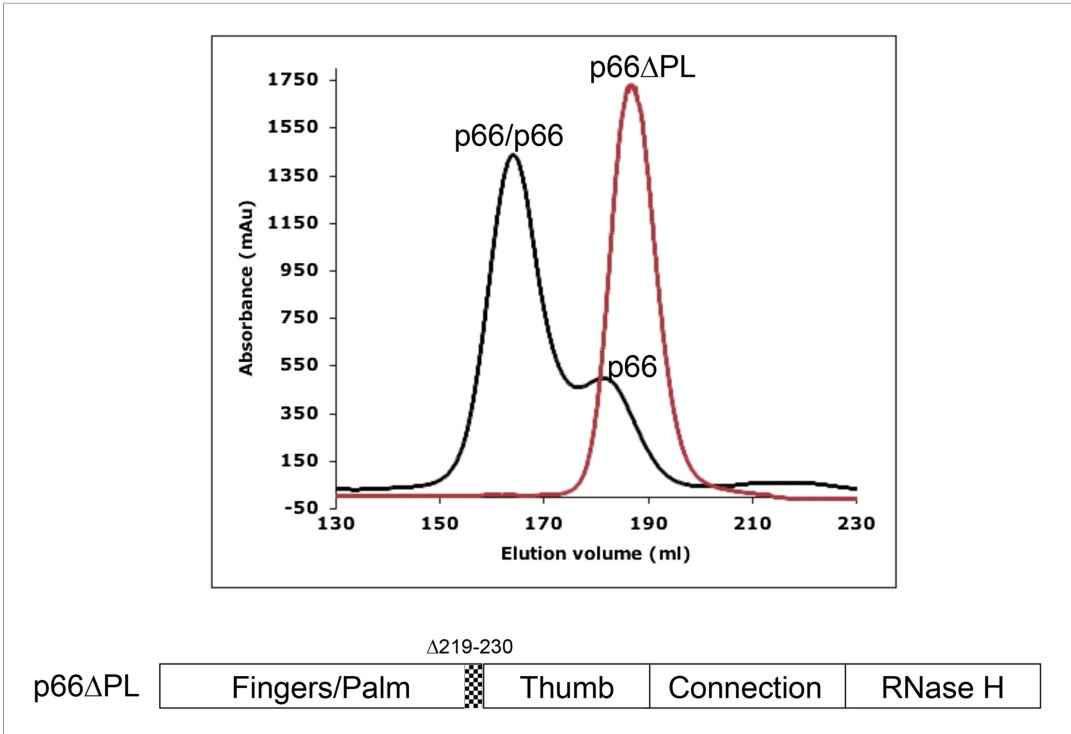

**Figure 2**. Effect of palm loop deletion on dimerization. Gel filtration chromatograms comparing p66 and p66ΔPL lacking palm loop residues 219–230. Chromatogram was obtained at 4°C on a HiLoad 26/60 superdex 200 column for p66 (black) and p66ΔPL (red) eluted with 50 mM Tris–HCl, pH 8.0, 200 mM NaCl. The palm loop deletion, developed to block isomerization, also fails to dimerize. The position of the deleted sequence in p66 is indicated at the bottom of the figure.

the monomer (p66M) or compact (p66C) species but strongly destabilizes the extended (p66E) form (*Zheng et al., 2014*).

The chromatograms shown in *Figure 2*, comparing the behavior of p66 and p66ΔPL on a size-exclusion column, demonstrate that under similar conditions, p66 exhibits a ~75/25 dimer/monomer ratio, while p66ΔPL fails to form any observable homodimer. This result follows directly from the conformational selection model, *Equation 1*, outlined above, since blocking isomerization will also prevent dimerization. Alternatively, if the conformational maturation of the polymerase domain occurred subsequent to dimer formation as described by *Equation 2*, we would expect to observe some dimer species. In principle, the loop deletion might interfere with dimerization by an undetermined mechanism; however, these residues are located just before the thumb domain and are not directly involved in the interface of the mature heterodimer. Thus, the behavior of p66ΔPL provides strong support for the conformational selection model.

## Intrinsic conformational preferences of the fingers/palm

The monomer structure provides an intuitive starting point for the spontaneous domain rearrangements that would be required for a conformational selection model. A structural comparison of the fingers/palm in an isolated construct (RT216, pdb: 1HAR) (*Unge et al., 1994*), the p51ΔPL monomer (pdb: 4KSE) (*Zheng et al., 2014*), the p51 and p66 subunits of RT (pdb: 1DLO) (*Hsiou et al., 1996*) reveals significant differences. This variation is most conveniently characterized by the angle between the approximately coplanar helices A (residues 28–43) in the fingers and F (residues 194–211) in the palm (*Figure 3*). In both the monomer and the p51 subunit of RT, this angle is ~45°. By comparison, in the isolated RT216 construct or the p66 subunit of RT, the angle is more obtuse, with values of 90°–100°. Importantly, in both of the structures with the more acute angle, there is a large interface between the fingers/palm and the connection domains. In contrast, for both of the

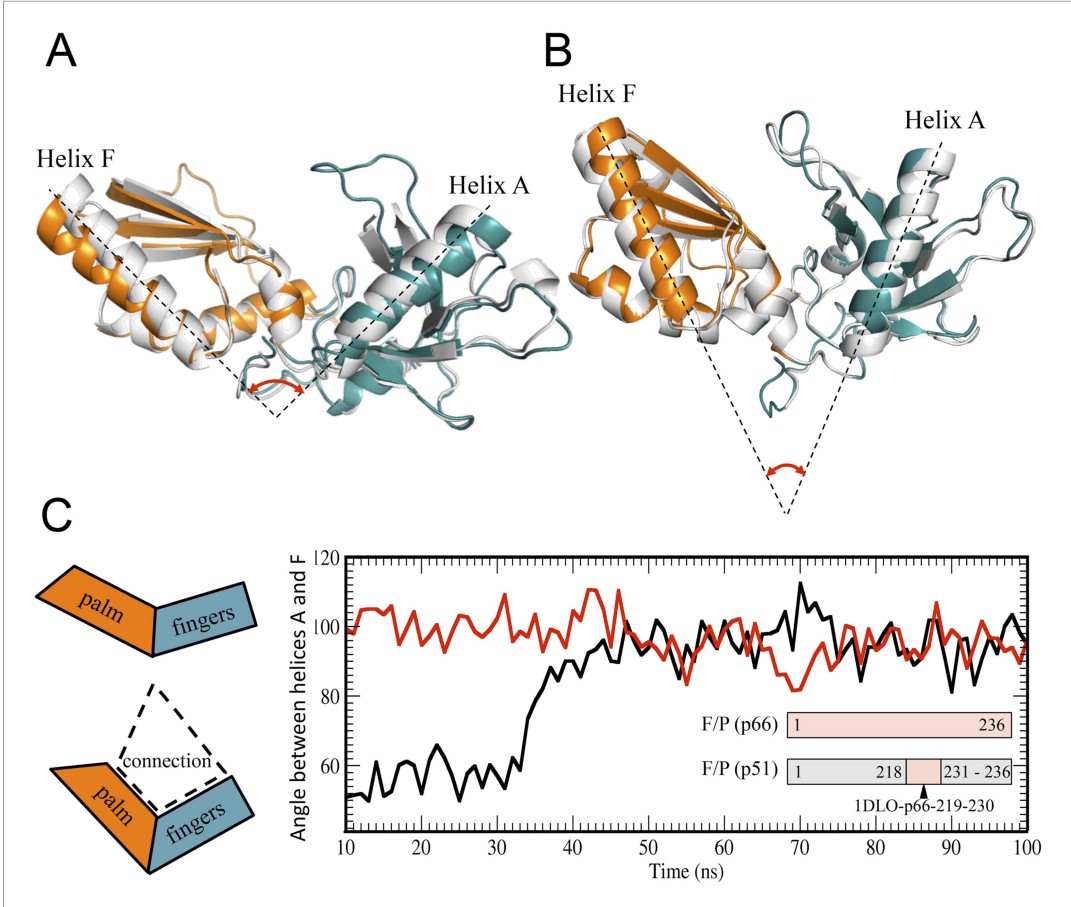

**Figure 3**. Alternative conformations and molecular dynamic simulations analysis of the fingers/palm subdomains. (**A**) Overlay of ribbon diagrams for fingers/palm residues 1–216 RT216 (pdb: 1HAR, gray) and in the p66 subunit of RT (pdb: 1DLO, fingers, teal; palm, orange). (**B**) Overlay of ribbon diagrams for the fingers/palm in the p51ΔPL monomer (pdb: 4KSE, gray) with the corresponding region of the p51 subunit of RT (pdb: 1DLO, fingers, teal; palm, orange). The fingers/palm angle defined by helices A and F is indicated, illustrating the more acute values for the monomer and the p51 subunit, compared with an isolated fingers/palm construct and the p66 subunit. (**C**) Time-dependent molecular dynamics simulations of the behavior of the $\alpha_{AF}$ angle for the fingers/palm starting with the p66 conformation (red) or with the p51 conformation (black). The simulations utilized residues 1–236 in the p66 and p51 subunits of RT (pdb: 1DLO) after removing all other domains at t = 0, and the missing palm loop residues in the p51 starting structure were introduced as indicated in 'Materials and methods'. Residues included in the simulations are defined in the inset. The cartoons on the left illustrate the starting fingers/palm conformations and the proposed role of the fingers/palm:connection interface in constraining the initial $\alpha_{AF}$ angle in the monomer and p51 structures.

The following figure supplement is available for figure 3:

**Figure supplement 1**. Additional simulations starting from the p51 monomer and from a structure that includes the p51 palm loop.

structures lacking this interface, the angle defined by helices A–F is much more open. This correlation suggests that the conformation with the more acute fingers/palm angle may be stabilized by the inter-domain interactions between the fingers/palm and the connection domains.

The above hypothesis was evaluated by performing molecular dynamics simulations on the isolated fingers/palm domains. Starting structures included residues 1–236 for the p66 and p51 subunits of apo RT (pdb: 1DLO). As discussed in 'Materials and methods', the missing p51 segment from 219–230 was modeled by introducing the corresponding segment from the p66 subunit. The fingers/palm $\alpha_{AF}$ angle was determined as a function of time after removal of all other domains. Simulations for the isolated fingers/palm starting with either the p66 or the p51 conformations are shown in

*Figure 3C*. As indicated in the figure, the more open conformation present in the p66 subunit (red line) is stable over the time period of the simulation. Alternatively, the simulation beginning with the fingers/palm in the p51 subunit (black line) indicates that between ~35 and 45 ns the $\alpha_{AF}$ angle undergoes a transition from its initial acute value to ~90˚. This result is consistent with an intrinsic preference for the open conformation observed in the crystal structure of the isolated fingers/palm construct, RT216 (pdb: 1HAR). Analogous simulations starting with the monomer structure (pdb: 4KSE) or with the p51 subunit of an RT-inhibitor complex containing the missing loop residues (pdb: 1S9E) produced qualitatively similar results, with the most significant variation related to the time at which the transition to the more extended conformation occurs (*Figure 3—figure supplement 1*).

The strategy of utilizing an isolated fingers/palm construct to reveal the intrinsic domain orientation preference is, however, subject to the limitation that inter-domain interactions involving the thumb and connection domains are omitted. Thus, although the final fingers/palm conformation produced by the simulations is similar to that observed in the p66 subunit of RT, the conformations of residues located at the domain boundaries do not agree with those in p66. The simulations are thus consistent with the general conclusion that the conformations of domain boundary residues depend on inter-domain interactions. This conclusion applies to residues in the palm loop, which fail to form strands of the larger β-sheet formed from palm and thumb residues.

Once the fingers/palm:connection subdomains have dissociated, palm loop residues can be recruited to cover exposed hydrophobic patches in the palm domain. The large fingers/palm: connection interface of ~1470 $\text{Å}^2$ in the p51 subunit of RT includes extensive hydrophobic contacts (*Ding et al., 1994*). In the monomer, these contacts include palm residues Leu100, Val106, Val108, Tyr181, Tyr188, and Leu234 (*Figure 4A*). In the active, p66*E* subunit of RT, this same group of hydrophobic residues in the palm domain interacts directly with residues from the palm loop (*Figure 4B*), which in the *E* conformation become part of the functionally important primer grip that positions the primer terminus for catalysis (*Ghosh et al., 1996*). Formation of alternate, intra-domain hydrophobic contacts by residues of the palm loop/primer grip can thus tend to interfere with re-association of the fingers/palm and connection domains, thereby enhancing the availability of the connection domain for intermolecular association with the monomer (*Figure 4C*). Further, these residues also form part of the binding site for non-nucleoside reverse transcriptase inhibitors (NNRTIs). This is a highly flexible region of the protein in which the NNRTI binding site is not identifiable in the absence of a bound inhibitor, and hence, is likely to be able to rapidly form intra-domain hydrophobic contacts that can inhibit connection domain re-association.

To summarize, structural data and molecular modeling simulations indicate that the more bent conformation of the fingers/palm present in the monomer structure does not represent a local minimum for the isolated fingers/palm, but a global minimum for the fingers/palm:connection complex. The fingers/palm apparently has an intrinsic preference for a more extended conformation that probably helps to promote dissociation of the fingers/palm:connection interface (*Figure 4C*). The inherent flexibility of the palm loop segment is expected to facilitate initial formation of intra-domain hydrophobic contacts that compete with inter-domain palm:connection interactions, reducing the tendency for re-association with the connection domain, and enhancing connection domain availability for dimerization.

## The initial p66/p66′ homodimer resembles the p51/p51′ homodimer

We previously presented data indicating that the p51/p51′ homodimer formed by the isolated polymerase domain exists as a conformational heterodimer (*Zheng et al., 2010*), a result consistent with its demonstrated polymerase activity (*Bavand et al., 1993*; *Dufour et al., 1998*). The scheme shown in *Figure 1* predicts that the initially formed p66/p66′ homodimer should resemble the p51/p51′ homodimer both of which lack the inter-subunit RH:thumb′ interface. To the extent that this analogy holds, the p51/p51′ homodimer should provide a stable model for the transiently formed initial p66/p66′ homodimer. A comparison of the $^1$H-$^{13}$C heteronuclear multiple-quantum correlation (HMQC) spectra obtained for the Ile-labeled p51 monomer and the p51/p51′ homodimer obtained under high salt conditions (*Figure 5A,B*) provides unequivocal evidence indicating conformational heterogeneity of the two subunits of the homodimer. In *Figure 5B* (see also *Figure 5—figure supplement 1*), the spectrum of the p51/p51′ homodimer (magenta) is overlaid with the spectra for p66-labeled RT (green) and p51-labeled RT (blue). The overlay demonstrates that the spectrum of p51/p51′ contains

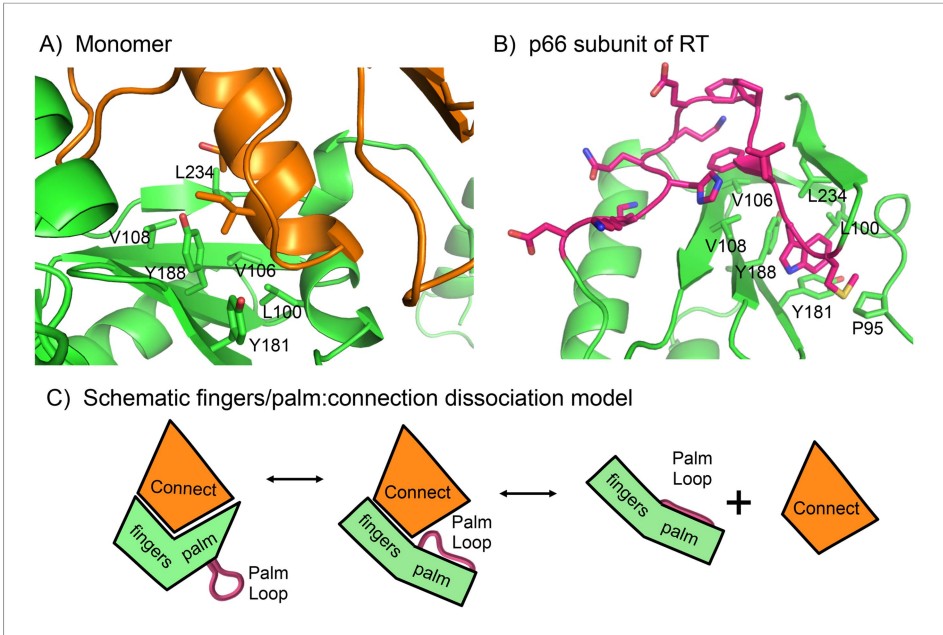

**Figure 4**. Role of the palm loop in isomerization of the polymerase domain. (**A**) Ribbon diagram of the p51ΔPL monomer (pdb: 4KSE, green) with the connection domain shown in orange. Several hydrophobic residues in the palm—Leu100, val106, Val108, Tyr181, Tyr188, and Leu234 that interact with the connection domain are annotated. (**B**) Ribbon diagram of the p66 subunit of RT (pdb: 1DLO) showing a portion of the fingers/palm domains (green) interacting with palm loop residues (219–230, magenta) of the palm domain. In the p66 subunit (*E* conformation), the palm loop becomes the primer grip and interacts with many of the same hydrophobic residues that interact with the connection domain in the monomer. (**C**) Schematic diagram illustrating how the intrinsic preference of the fingers palm for a more open conformation facilitates disruption of the fingers/palm:connection interface and repositioning of the palm loop.

multiple resonances that are nearly coincident with resonances from both the p66 and the p51 subunits of RT. Thus, the p51/p51′ homodimer exists as a conformational heterodimer that is structurally similar to the RT heterodimer and contains *both E*-like and *C*-like conformations.

A more complete analysis of the $^1$H-$^{13}$C HMQC spectrum of the Ile-labeled p51/p51′ dimer indicates that it contains resonances that are in close agreement with resonances from the fingers, palm, thumb, and connection domain of the p66 subunit, while lacking resonances attributable to the RH domain. We, thus, conclude that the conformation of the p51 subunit of the p51/p51′ homodimer can be characterized as adopting a p51*E*-like conformation, that is, similar to the p66 conformation of the heterodimer but lacking an RH domain. In contrast, the conformation of the p51′ subunit of the homodimer is more difficult to characterize. In some cases, for example, Ile202′ and Ile47′, the resonances are in close agreement with those of the p51*C* subunit of the RT heterodimer (blue spectrum), while in other cases, for example, Ile274′, Ile329′, and Ile375′, resonances near the positions expected for p51*C* are not observed (*Figure 5B*). The resonance of Ile274′ from the thumb′ domain is at the position of p51*M* rather than p51*C*, and the connection′ Ile329′ and Ile375′ resonances are not readily observed, as is the case with the monomer. This behavior indicates that formation of the p51/p51′ dimer leads to shifts in the fingers′/palm′ that are consistent with dimer formation, while several of the p51′ thumb′ and connection′ domain resonances more closely approximate the pattern of the p51 monomer. We conclude that the NMR data support a homodimer model in which the p51 subunit approximates the p66 RT subunit without an RH domain, while the p51′ subunit conformation approximates that of the p51 monomer that includes a disordered thumb′ and disordered C-terminal αM′ residues (*Figure 5B* schematic and *Figure 1—figure supplement 1C*). Apparently, the interactions between helix αM′ and the thumb′ are insufficient to stabilize a p51*C* conformation similar to that observed in the heterodimer, indicating

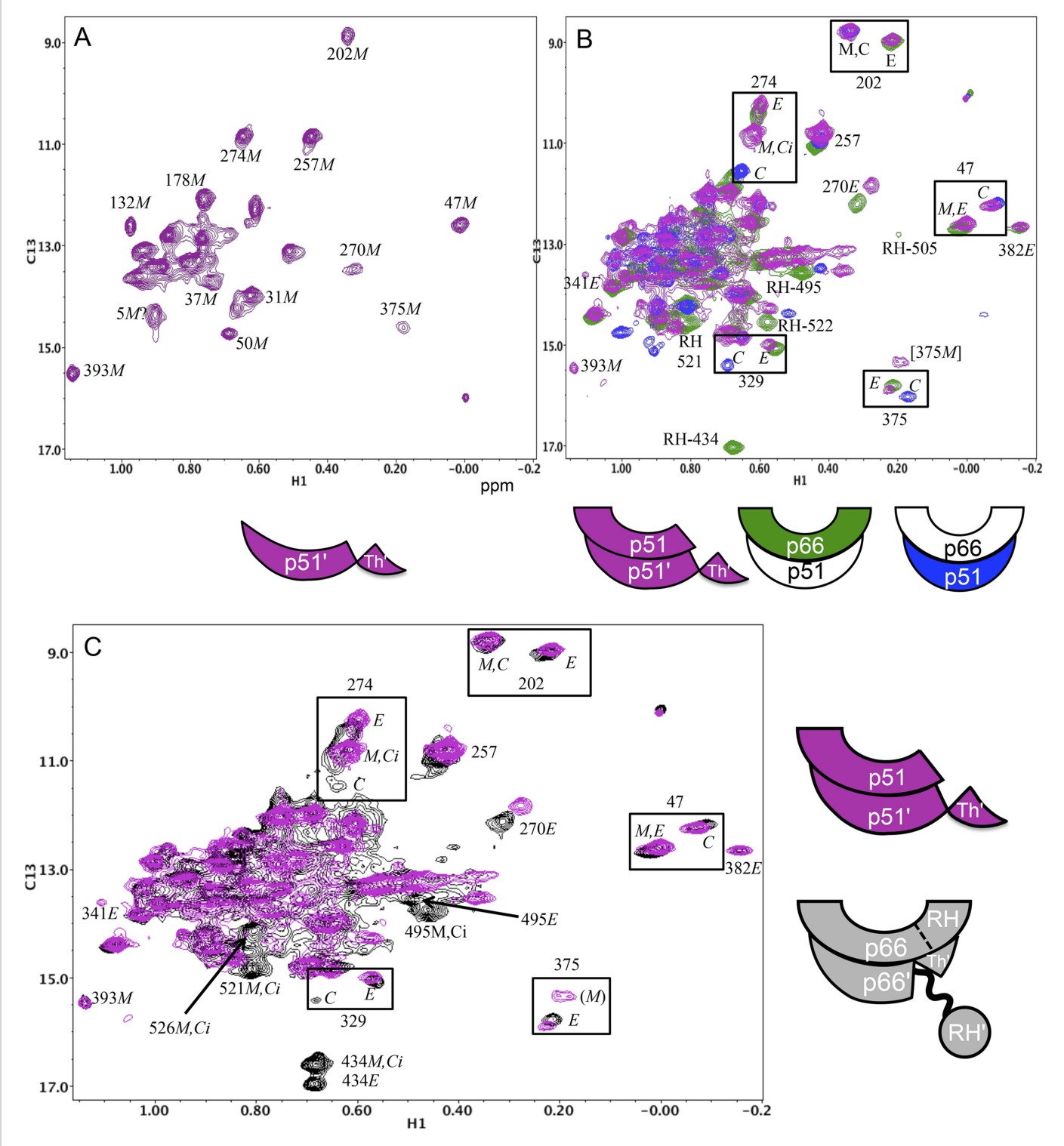

**Figure 5**. Spectral comparisons of p51/p51' and p66/p66' homodimers. (**A**) $^1$H-$^{13}$C heteronuclear multiple-quantum correlation (HMQC) spectrum of the [$^{13}$CH$_3$-Ile]p51 monomer. (**B**) Overlaid HMQC spectra of the [$^{13}$CH$_3$-Ile]p51/[$^{13}$CH$_3$-Ile]p51' homodimer with the spectra for [$^{13}$CH$_3$-Ile]p66/p51 (green) and p66/[$^{13}$CH$_3$-Ile]p51 (blue). We note the absence of homodimer resonances that overlay the resolved RH domain resonances in p66-labeled RT. (**C**) Overlaid HMQC spectra for the labeled p51 homodimer and the Ile-labeled p66/p66' homodimer obtained during the first 5.5-hr accumulation period after initiation of dimerization. The p66 homodimerization studies were performed in 25 mM Tris-HCl-d11 in D$_2$O, pD = 7.51, 100 mM KCl, 0.02% NaN$_3$. In order to stabilize the p51/p51' homodimer, it was necessary to use a high salt buffer containing 800 mM KCl and 20 mM MgCl$_2$ in addition to the other

*Figure 5. continued on next page*

*Figure 5. Continued*

components. The labeling pattern corresponds to the color coding in the cartoons near each spectrum, with white indicating an unlabeled subunit. The assignment in parenthesis is considered tentative. The RH, RH', and Th' labels in the cartoon indicate the RNase H domain in the p66 subunit, the RNase H domain in the p66' subunit, and the Thumb' domain in the p51' or p66' subunits.

The following figure supplements are available for figure 5:

**Figure supplement 1**. Spectral comparison of the p51/p51' and initial p66/p66' homodimers.

**Figure supplement 2**. Spectral comparison of the initial p66/p66' homodimer with the selectively-labeled subunits of RT.

**Figure supplement 3**. Assignments of connection domain resonances in the p66 subunit of RT.

**Figure supplement 4**. Table of mutated residues.

**Figure supplement 5**. Resonance perturbations in [$^{13}CH_3$-Ile]p66(I341V)/p51.

**Figure supplement 6**. Resonance perturbations in [$^{13}CH_3$-Ile]p66(I382V)/p51.

**Figure supplement 7**. Resonance perturbations in [$^{13}CH_3$-Ile]p66(I270V)/p51.

**Figure supplement 8**. Resonance perturbations in [$^{13}CH_3$-Ile]p66(Y342H)/p51.

**Figure supplement 9**. Resonance perturbations in [$^{13}CH_3$-Ile]p66(I167V)/p51.

**Figure supplement 10**. Resonance perturbations in [$^{13}CH_3$-Ile]p66(I526V)/p51.

**Figure supplement 11**. Resonance perturbations in [$^{13}CH_3$-Ile]p66(I522V)/p51.

**Figure supplement 12**. Resonance perturbations in [$^{13}CH_3$-Ile]p66(H361Y)/p51.

**Figure supplement 13**. Resonance perturbations in [$^{13}CH_3$-Ile]p66(S379C)/p51.

**Figure supplement 14**. Resonance perturbations in [$^{13}CH_3$-Ile]p66(I375V)/p51.

**Figure supplement 15**. $^1$H-$^{13}$C HMQC spectrum of [$^{13}CH_3$-Ile]p66(I375V)/ [$^{13}CH_3$-Ile]p66(I375V)' mature homodimer.

that the additional interactions with the RH domain are required for this conformation to be significantly populated.

Extensive similarities are observed in an overlay of the $^1$H-$^{13}$C HMQC spectrum of the Ile-labeled p51/p51' with the spectrum of the p66/p66' homodimer obtained during the first 5.5-hr accumulation period after initiation of dimerization (*Figure 5C*). The p66/p66' spectrum further demonstrates even closer agreement with the spectra of the RT heterodimer (*Figure 5—figure supplement 2*), demonstrating the presence of *E*-like and *C*-like conformers. This result is in direct conflict with the model recently proposed by *Sharaf et al. (2014)* in which the initial p66 homodimer observed by NMR exists as a conformationally symmetric homodimer. Consistent with our previous study (*Zheng et al., 2014*), resolved RH domain resonances indicate that the early p66/p66' homodimer contains two-folded RH domains, one of which exhibits a shift pattern similar to that of the isolated subunit. This behavior is most readily observed for the isolated Ile434 resonances, and considered in greater detail in the following sections.

## Formation of the inter-subunit RH:thumb' interface

As outlined in *Figure 1* (see also *Figure 1—figure supplement 1*), the initially formed homodimer lacks an RH:thumb' interface. In order to more directly address the question of *when* this interface is formed, it was first necessary to determine how interface formation affects the isoleucine resonances

in the p66 RH domain, and particularly the shift of Ile434, which is located in p66 RH near the RH: thumb' interface. The strategy presented below compares the Ile shifts in the p66 subunit of the wt RT heterodimer with the shifts in a mutant heterodimer containing a p51 thumb' mutation positioned at the interface with RH. Specifically, residue Leu289 on the p51 subunit interacts with a hydrophobic pocket on the p66 RH domain, so the non-conservative p51(L289K) mutant should significantly disrupt the structure of this interface in the p66/p51(L289K) heterodimer. A comparison of the NMR spectra obtained for [$^{13}$CH$_3$-Ile]p66/p51(L289K) with the spectrum obtained for the non-mutated protein will reveal the shift perturbations that result from interface formation.

In *Figure 6A*, we compare the $^1$H-$^{13}$C HMQC spectrum of the p51-mutated, p66-labeled heterodimer, [$^{13}$CH$_3$-Ile]p66/p51(L289K), with the spectrum obtained for the p66-labeled heterodimer lacking the p51 thumb mutation. In order to overcome the reduced tendency of mutated p51 to dimerize (*Goel et al., 1993*; *Zheng et al., 2010*), we utilized a twofold excess of unlabeled p51(L289K) to enhance dimer formation with labeled p66. The spectra in *Figure 6A* demonstrate that this strategy was successful; the resonance pattern observed for [$^{13}$CH$_3$-Ile]p66/p51(L289K) is qualitatively similar to that obtained for wt RT labeled in the p66 subunit (*Zheng et al., 2014*), while resonances with shifts that are characteristic of the p66 monomer, for example, Ile393M and Ile274M are very weak. In addition, resonances characteristic of the p66C conformation of the homodimer, for example, Ile202C and Ile47C, which would be present if p66/p66' containing labeled Ile in both subunits was present, are weak or absent. Thus, nearly all of the label has ended up in the p66E subunit of the RT heterodimer, [$^{13}$CH$_3$-Ile]p66/p51(L289K), rather than in a p66 monomer or a p66/p66' homodimer.

Disruption of the p51 thumb':p66 RH interface by the p51(L289K) mutation alters many of the shifts within the p66 RH domain. The shift differences of the resolved Ile434 and Ile495 resonances that are characteristic of the RT-incorporated RH domain are eliminated. Thus, the $^{13}$C shifts of Ile434 (16.6 ppm) and Ile495 (13.7 ppm) observed in [$^{13}$CH$_3$-Ile]p66/p51(L289K) are similar to the values in the isolated RH domain, but differ from the values of 17.0 and 13.6 ppm observed in the wt RT heterodimer. The shift differences summarized above, thus, allow us to determine at what point in the maturation process the RH:thumb' interface is formed. A $^{13}$C shift of 17.0 ppm for the Ile434 resonance indicates that the inter-subunit RH:thumb' interface has formed, while a shift of ~16.6 ppm, similar to that of the isolated RH domain and also observed in the mutant heterodimer discussed above, indicates that this interface is not present or not well-formed. Since we observe strong intensity for the Ile434 resonance at 17.0 ppm during the first 5.5-hr accumulation period after conditions favoring the homodimer are introduced (*Figure 5C*), we conclude that the thumb':RH interface has largely been formed during this initial period. A similar conclusion follows from analysis of the shifts of the Ile495 resonance.

Interestingly, the $^1$H-$^{13}$C HMQC spectrum of [$^{13}$CH$_3$-Ile]p66/p51(L289K) exhibits multiple additional shift perturbations of other RH and connection domain resonances. Methyl resonances of Ile522, located at the connection:RH interface, resonances of Ile329, Ile375, and Ile382 located within the p66 connection domain experience significant broadening, and the Ile341 resonance is shifted. Locations of the p66 Ile residues exhibiting these perturbations are illustrated in *Figure 6B*. Since as shown above, the L289K' perturbation is insufficient to prevent heterodimer formation, we conclude that perturbation of the thumb':RH interface with the L289K' mutation introduces additional perturbations that extend into the RH and connection domains of p66. These observations highlight the cooperative nature of interface formation in RT.

## Formation of the p66:p66' interface

The model shown in *Figure 1* describes a conformational selection process in which the predominant monomer 'selects' a structurally isomerized p66 molecule in a rare, p66E$_i$ conformation as its initial binding partner. Initial dimer formation probably involves non-specific hydrophobic contacts between the connection domains. A comparison of the connection domains in the monomer, the p66 subunit, and the p51 subunit of RT reveals significant structural variations, particularly in regions involved in interface formation (*Figure 1—figure supplement 2*), so that a simple rearrangement of domain positions is insufficient to result in formation of an interface similar to that of the mature heterodimer; additional conformational changes within the connection domain are also required. This requirement is most clearly apparent from an overlay of the connection domain in the monomer with the connection domain on the p66 subunit of the heterodimer (*Figure 1—figure supplement 2*). Among the various structural changes that must occur, straightening of helix αL in the *E* conformation alters

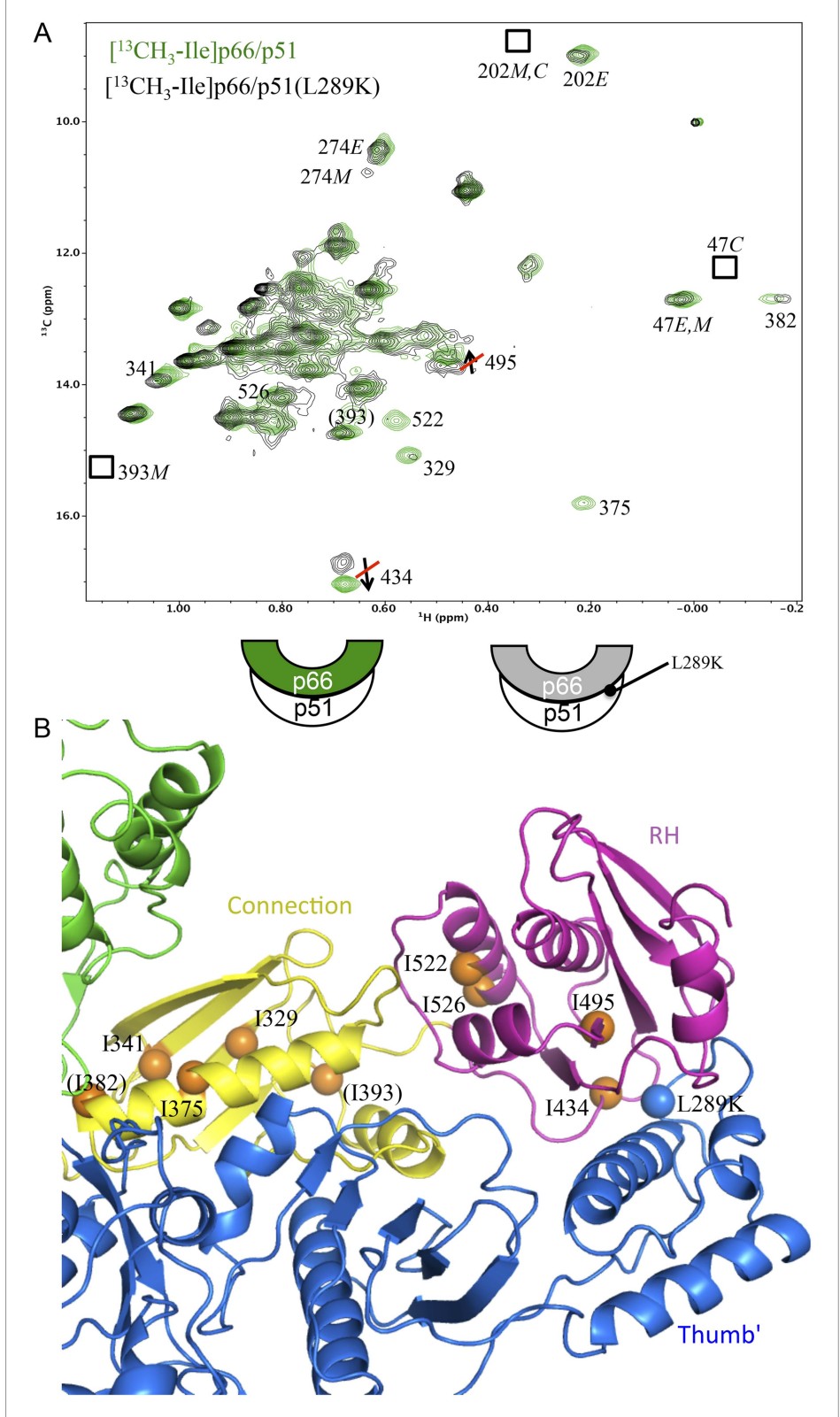

**Figure 6**. Effect of a p51 thumb' domain mutation on the Ile methyl resonances in the p66 subunit of RT. (**A**) Overlay of the $^1$H-$^{13}$C HMQC spectrum of [$^{13}$CH$_3$-Ile]p66/p51 (green) and [$^{13}$CH$_3$-Ile]p66/p51(L289K) (black). Most of the features of the spectrum are preserved, consistent with the formation of a stable heterodimer. The labeling pattern corresponds to the color-coding in the cartoons below the spectrum. (**B**) Ribbon diagram illustrating the relative

*Figure 6. Continued*

position of the mutated residue (blue sphere) and perturbed resonances in the RH and connection domains of p66 (orange spheres). Color coding: p51 (blue); p66 RH domain (magenta); p66 connection domain (yellow), p66 fingers/palm and thumb domains (green).

multiple intra- and inter-domain contacts facilitating inter-subunit interface formation. Consequently, initial dimer formation involving the connection domains prior to this conformational change must include many non-specific hydrophobic contacts.

A comparison of the $^1$H-$^{13}$C HMQC spectra obtained for the initial [$^{13}$CH$_3$-Ile]p66/[$^{13}$CH$_3$-Ile]p66′ homodimer with the spectra obtained for the subunit-labeled heterodimer (*Figure 5—figure supplement 2*) indicates that all of the resolved Ile resonances of residues in the extended p66 conformation, for example, Ile329*E*, Ile341*E*, Ile375*E*, and Ile382*E* are readily observed. These resonances characteristic of the connection domain in the *E* conformation are not present in the monomer or in the spectra of labeled p66*C*. Thus, the conformational changes required to alter the connection domain from its monomer to its p66*E* conformation have largely been completed during the first accumulation period. In addition to the connection domain resonances, resonances attributed to residues in the fingers (Ile47), palm (Ile202), thumb (Ile274), and RH domain (Ile434) also are in agreement with resonances in the p66-labeled RT spectrum (*Figure 5—figure supplement 2*). These observations are consistent with the results summarized in the previous section, indicating that the RH:thumb′ interface has largely been formed during the first accumulation period.

Time-dependent intensity data for connection domain resonances assigned to Ile329*E*, Ile375*E*, and Ile382*E*, summarized in *Table 1*, give time constants of ~2–3 hr, shorter than the 5.5-hr accumulation used for the first spectrum. Thus, the p66 subunit has evolved from an initial conformation involving non-specific hydrophobic contacts to a form that closely approximates its mature, p66*E* conformation during the initial accumulation period.

In contrast with the behavior of the p66 subunit summarized above, resonances arising from p66′ support a more complex interpretation. Fingers/palm resonances Ile47′ and Ile202′ are at the expected Ile47*C* and Ile202*C* positions characteristic of the mature dimer. For Ile47′, there is a significant shift difference between the monomer and the dimer, so that this result supports the conclusion that the region of the interface near Ile47′ is structurally similar to that of the mature heterodimer. In contrast, connection′ domain resonances Ile329′ and Ile375′ are weak or absent, that is, more similar to their behavior in the monomer. We have assigned two resonances to Ile274′: a more intense peak with a shift close to the monomer (Ile274*C$_i$*) and a second weaker peak with a shift close to position of the mature heterodimer (Ile274*C*). Based on intensity comparisons with the Ile393*M* resonance, the Ile274*Ci* peak is attributed mostly to an immature dimer species with a monomer-like shift, while the weaker Ile274*C* resonance is attributed to the p66′ subunit of the conformationally mature p66/p66′ homodimer.

Importantly, the evidence outlined in the previous section indicates that the thumb′:RH interface is largely formed during the first accumulation period; however, the Ile274′ resonance is mostly at the monomer position in the first p66/p66′ spectrum. This difference may indicate that the base of the thumb′ undergoes a slow conformational maturation process that is separate from formation of the thumb′:RH interface. Alternatively, Ile274′ is sufficiently close to the connection′ domain so that its time-dependent shift behavior may be sensitive to changes that are occurring in the connection′ domain, and particularly to the formation of helix αM′.

Ile residues located at or near the subunit interface include: Ile159, Ile380, Ile382, Ile411, and Ile542 on p66, and Ile135 on p66′. However, due to broadening and/or resolution limitations, only Ile382 provides a useful probe for dimer formation (*Figures 5B,C and 6*). In the heterodimer structure, pdb: 1DLO (*Hsiou et al., 1996*), the Ile382 δ-methyl is positioned 5.6 Å from the sidechain carbonyl oxygen of Asn136 on the p51 subunit. The Asn136 residue on p51 and the loop containing it have been shown to play an important role in dimerization (*Balzarini et al., 2005*; *Mulky et al., 2007*; *Upadhyay et al., 2010*). Based on the behavior of the Ile382*E* resonance, this interface is formed at a sufficiently early stage so that it is largely present during the first 2D $^1$H-$^{13}$C HMQC accumulation period of 5.5 hr. Analysis of the time-dependent data gave a time constant of 2.4 ± 0.2 hr (*Table 1*), consistent with a relatively early formation of this portion of the interface involving the connection and fingers′

**Table 1**. Apparent time constants—homodimerization study

| Residue | Mean ± S.E.* |
|---|---|
| 329E | 3.3 ± 0.5 |
| 375E | 2.6 ± 0.7 |
| 382E | 2.4 ± 0.2 |
| 329C | 5.8 ± 0.3 |
| 375C[b] | 5.9 |
| 274C | 8.9 ± 0.6 |

*Fitted parameters are averages ±standard error for three separate studies. [b]For Ile375C, one data set was obscured by a spectral artifact, so the tabulated value is the average of two measurements. Illustrative data fits of individual data sets are shown in **Figure 7—figure supplement 1**.

domains. This conclusion also follows from the time-dependent behavior of the Ile47' resonance discussed above.

In summary, dimerization is occurring on a scale too rapid for direct NMR observation, however, comparisons of resonance shifts with values in the monomer and heterodimer, as well as structural comparisons with the monomer, indicate that several conformational steps are largely completed during the initial accumulation period. These include maturation of the dimer interface so that the p66 connection domain matures from its monomer to its extended (*E*) conformation and formation of the RH:thumb' domain interface. Maturation of the connection' proceeds on a slower time scale.

## RH' unfolding is coupled with connection' maturation

We previously proposed that the supernumerary RH' domain initially present in the p66' subunit of the homodimer is destabilized and unfolds as a result of transfer of residues near Tyr427' that develop into helix αM' in the connection' domain of the mature p66' subunit. This model was supported by the decay of several resonances that could be assigned specifically to the RH' domain (*Zheng et al., 2014*). The more complete assignments of the connection domain included with the present study (*Figure 5—figure supplements 3–15*) provide further substantiation of this hypothesis. The Ile329 and Ile375 resonances are particularly useful for analysis of connection domain conformational processes since they are well resolved and give unique signals characteristic of the *E* and *C* conformations. These resonances are also not readily observed in the monomers, probably as a result of exchange broadening (although a broad resonance in the general region of Ile375 may correspond to this residue). The I375V mutation eliminates both the Ile375 and Ile329 resonances as a consequence of the proximity of these two residues ($\delta CH_3$(Ile329)-$\delta CH_3$(Ile375) = 3.4 Å in 1DLO) (*Figure 5—figure supplement 14*). The spectrum of the mature p66(I375V)/p66'(I375'V) homodimer (*Figure 5—figure supplement 15*) shows the same two missing resonances arising from the p66 subunit and also identifies two additional perturbed resonances that we assign to the corresponding residues in the p66' subunit of the homodimer.

*Figure 7A* shows four ${}^1H$-${}^{13}C$ HMQC spectra of Ile-labeled p66 at successive 5.5-hr time periods after dimerization conditions are introduced, for a spectral region containing the Ile329, Ile375, and Ile434 resonances. Consistent with the behavior summarized above, the three resonances assigned to residues in the p66 subunit: Ile329*E*, Ile375*E*, and Ile434*E* are approaching their equilibrium intensities during the first NMR accumulation. During the subsequent accumulation periods, the Ile434*C* resonance, which contains contributions from both the p66' subunit of the homodimer and from the overlapping Ile434*M* resonance of the monomer, decays almost completely. The Ile329*C* and Ile375*C* resonances arising from the connection' domain of the p66' subunit of the homodimer show gradual intensity gains over this same time period. We attribute these changes to the simultaneous destabilization of RH' and the conformational maturation of the connection' as residues derived from RH' are incorporated into helix αM'. The temporal linkage of these events is consistent with a model in which they are functionally coupled processes. These occur on a much slower time scale than the conformational processes described in the previous section that include isomerization of the monomer to an *E*-type conformation, initial formation of the immature homodimer, and formation of the RH:thumb' interface.

Three resonances are assigned to Ile274, located near the base of the thumb (*Figure 7B*). As indicated in *Figure 5*, the positions approximate the shifts characteristic of the *M, C,* and *E* conformations. The behavior of Ile274*E* is similar to the other resonances assigned to the *E* conformer, with the intensity nearing its limiting value during the first accumulation period. The intensity of the $C_i$ resonance, closest to the monomer position, decays on a slow time scale, while the intensity of the

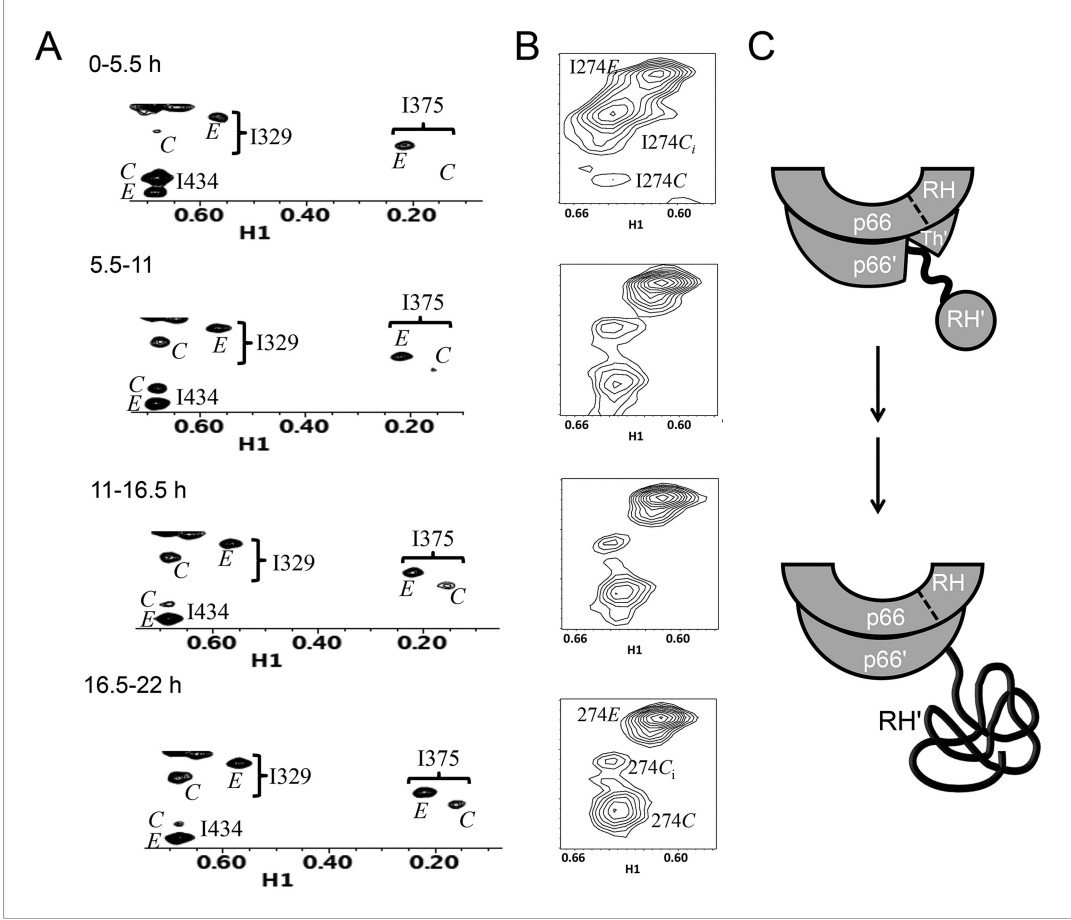

**Figure 7**. Slow time-dependent changes of connection and RH domain resonances. (**A**) An expanded spectral region of the [$^{13}$CH$_3$-Ile]p66/[$^{13}$CH$_3$-Ile]p66' homodimer obtained at successive time intervals after introduction of conditions favoring dimerization. The selected region includes connection and connection' Ile329 and Ile375 resonances as well as RH and RH' Ile434 resonances. (**B**) The time-dependent changes of the Ile274 resonances during the same time period. (**C**) A schematic diagram illustrating the conformational changes in the connection' and RH' domains that are related to the observed resonance changes. The labeled subunits are indicated in gray. The RH, RH', and Th' labels in the cartoon indicate the RNase H domain in the p66 subunit, the RNase H domain in the p66' subunit, and the Thumb' domain in the p66' subunit. Data supporting the assignments of the connection and connection' domain Ile329 and Ile375 resonances are presented in *Figure 5—figure supplements 4, 13, 14, and 15*. Dimerization was initiated at t = 0, and the spectra were obtained at 35°C.
The following figure supplement is available for figure 7:

**Figure supplement 1**. Illustrative fits of time-dependent intensity data.

Ile274$C$ resonance grows over a similar time period. The Ile374 $C_i$ resonance is attributed to the initially formed homodimer rather than to the monomer (*Figure 1—figure supplement 1D*), since it is much greater than that of the other monomer resonances, for example, Ile393$M$. This behavior indicates that either the base of the thumb' is experiencing a slow conformational maturation or, more probably, that Ile274' is sufficiently close to the connection' domain, and particularly to αM', so that the resonance is sensitive to changes occurring in the nearby domain.

Time constants determined from the time-dependent intensities of the connection domain Ile329 and Ile375 resonances are summarized in *Table 1*, and representative data fits are shown in *Figure 7—figure supplement 1*. As noted above, the Ile329$E$ and 375$E$ resonances increase with time constants that are shorter than the length of the first 5.5-hr accumulation period, consistent with

the model of *Figure 1* in which isomerization of the monomer to the extended *E* conformation is the initial step. Alternatively, the Ile329*C* and Ile375*C* resonances in the connection' domain increase with slower time constants of ~6 hr (*Table 1*) that are similar to those reported previously for the decay of the RH' Ile434*C*, Ile495*C*, and Ile521*C* resonances (*Zheng et al., 2014*), consistent with the coupled residue transfer model outlined above. The slow forming Ile274*C* resonance attributed to the p66' thumb' exhibited a somewhat slower time constant of almost 9 hr (*Table 1*). This may correspond to an even slower maturation step; however, there are insufficient data to further develop a more specific hypothesis.

## Conformationally selective labeling and kinetic perturbations with a deletion mutant

One of the difficulties of analyzing homodimer maturation by NMR is the presence of isotopic labels in both subunits. Based on the ability of the palm loop deletion to block formation of the p66*E* conformation, we performed a time-dependent dimerization study of [$^{13}$CH$_3$-Ile]p66ΔPL in the presence of a twofold molar excess of unlabeled p66 in order to facilitate complete conversion of the p66ΔPL to the dimer form (*Figure 8*). The time-dependent spectral changes were qualitatively similar to those observed in the homodimerization study (*Figure 8—figure supplement 1*). The region of the $^1$H-$^{13}$C HMQC spectrum shown in *Figure 7* that includes several Ile resonances arising from the connection' and RH' domains shows the same time-dependent decay of the RH' Ile434*C* and Ile521*C* resonances in parallel with increases in the intensities of the connection' Ile329*C* and Ile375*C* resonances in the study using the palm loop deletion (*Figure 8*). Thus, as in the homodimerization study, the data demonstrate that formation of the connection' domain is temporally correlated with the disappearance of resonances characteristic of the folded RH'. This observation further supports the maturation of the connection' domain at the expense of the RH' domain. In these studies, none of the resonances uniquely attributed to the *E* conformation was observed, indicating that the p66ΔPL subunit of the pseudo-homodimer does not adopt the *E* conformation to any significant extent. Thus, consistent with expectations based on the behavior illustrated in *Figure 2*, the labeled p66ΔPL is unable to form a homodimer or to dimerize with the p66 monomer by adopting the extended (*E*) conformation.

Despite qualitative similarity with the homodimerization study, the kinetic behavior exhibits significant differences (*Table 2*). Most importantly, the fraction of p66ΔPL initially in the monomer form is greater than that observed in the homodimerization study, the decay of the monomer resonances is slower, and the dimerization is incomplete, reaching only 80–90 % based on comparisons of the intensities of multiple resonances (*Figure 8—figure supplement 2*). These kinetic differences can be interpreted within the context of the conformational maturation model (*Equation 1*) as resulting from competition between the unlabeled p66 monomer and p66ΔPL for the rare p66*E* conformation that is only formed by isomerization of p66. Note that all of the steps except perhaps for the final RH' unfolding are expected to be fully reversible, so that the NMR observations represent average populations of the observed species that indicate the conformational mixture present during each accumulation period. Unfolding of the RH' domain on the p66' subunit of p66/p66' will further deplete the pool of p66 available to form the p66*E* species.

The presence of a significant monomer concentration affects the kinetic analysis in two ways: 1) there is a gradual contribution to the intensities of all labeled dimer resonances as the monomer is converted to dimer and its various incompletely matured forms, and 2) in some cases, for example, the RH' resonances, the monomer resonances overlap those in the dimer. Due to the more significant effects of the monomer in this study, we have not attempted to introduce a monomer correction, as was done in our previous analysis (*Zheng et al., 2014*), and instead just fit the data to the simplest mathematical models that provided reasonable approximations. The results, summarized in *Table 2*, indicate that: 1) the initial monomer concentration has decreased by ~ 40% after which it decays with a time constant of ~10 hr based on Ile393*M* and Ile47*M* peaks, reaching a limiting level of ~20% of the total. 2) the RH' Ile434', Ile495', and Ile521' resonances all decay with similar apparent time constants with a mean value of 10.7 hr, and the connection' Ile329'; and Ile375' resonances increase with similar time constants, kinetically linking these two processes. 3) the Ile47' resonance intensity is divided between monomer and dimer species, Ile47*M* and Ile47*C*, so that the time-dependent behavior results from the monomer→ dimer conversion (*Figure 8—figure supplement 2*). In the initial spectra, the dimer species accounts for 30–40 % of the total, after which it increases with a time constant of

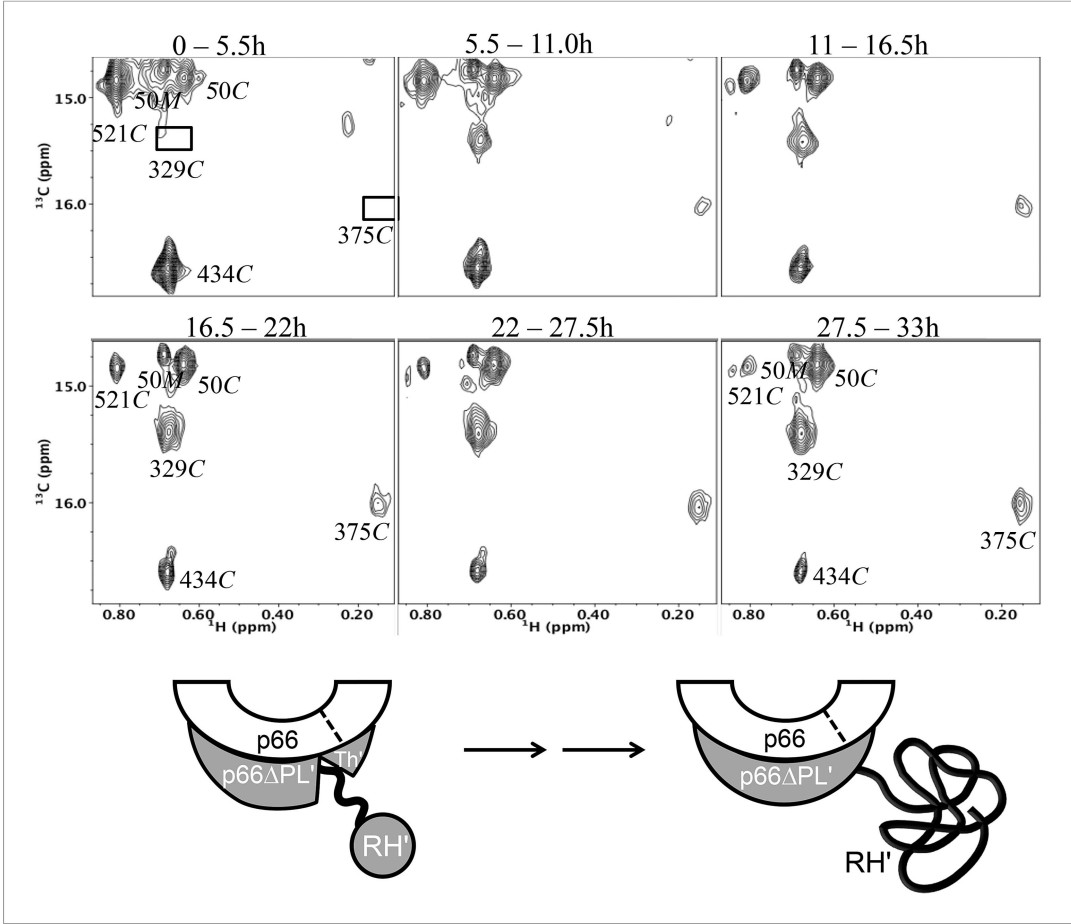

**Figure 8**. Dimerization of [$^{13}$CH$_3$-Ile]p66ΔPL with unlabeled p66. Time-dependent changes are shown for a region of the $^1$H-$^{13}$C HMQC spectrum covering a similar spectral region to that shown in **Figure 7**. All resonances are attributed to the M or C species; the labeled Ile50M resonance as well as the Ile434C and 521C resonances arising from the labeled RH' decrease as the RH' domain unfolds, while the connection' 329C and 375C resonances increase as the connection' domain matures. The schematic diagram at the bottom illustrates the subunit-selective labeling pattern and the proposed conformational changes that are inferred from the behavior of the resonances. The labeled subunit is indicated in gray. Each spectrum corresponds to a 5.5-hr accumulation period at the time periods indicated. Dimerization was initiated at t = 0, and the spectra were obtained at 35°C.

The following figure supplements are available for figure 8:

**Figure supplement 1**. Time-dependent HMQC spectra for dimerization of p66ΔPL with excess, unlabeled p66 showing all Ile δ-methyl resonances.

**Figure supplement 2**. Time-dependent intensity data for monomer decay.

**Figure supplement 3**. Time-dependent decay of RH' resonances.

**Figure supplement 4**. Time-dependent growth of connection' resonances.

~9 hr, leveling off at about 80% of the total intensity (**Table 2**). 4). All of these rates are longer than the 5.9 hr time constant observed for maturation of the connection' Ile329' and Ile375' resonances in the homodimerization study (**Table 1**).

The behavior summarized above, particularly for the Ile47' resonances, suggests that dimerization of labeled p66ΔPL with p66E is initially a rapid process but slows down significantly possibly as the pool of p66E monomer becomes depleted due to dimer formation. Subsequent dimerization of p66ΔPL may

**Table 2**. Apparent time constants—p66 + [$^{13}$CH$_3$-Ile]p66$\Delta$PL

| Residue | Mean ± S.E.* |
| --- | --- |
| 393*M* | 9.3 ± 1.4 |
| 274*M*/C$_i$ | 8.4 ± 0.5 |
| 47*M* | 9.1 ± 1.7 |
| Mean monomer decay TC | 9.0 ± 0.6 |
| 47*C* | 8.8 ± 1.2 |
| 434*C*† | 10.0 ± 0.2 |
| 495*C*† | 10.4 ± 0.4 |
| 521*C*† | 11.6 ± 0.3 |
| Mean RH resonance decay TC | 10.7 ± 0.3 |
| 329*C* | 10.3 ± 1.6 |
| 375*C* | 11.2 ± 2.5 |
| Mean connection' growth TC | 10.8 ± 1.3 |

*Errors determined as in **Table 1**. Each value represents the mean of three separate studies.
†Resonances 434*C*, 495*C*, and 521*C* also contain contributions from overlapping monomer peaks, and no attempt has been made to correct for this overlap. Similarly, the resonance labeled 274*M*/C$_i$ contains contributions from both the monomer and the initially formed dimer, so that the decay results from both dimerization and conformational maturation of the dimer. Illustrative data fits are shown in **Figure 8—figure supplements 2–4**.
RH: ribonuclease H; TC = time constant.

require release of p66 monomers from p66/p66′ dimers at various stages of conformational maturation, until most of the p66 and p66ΔPL have formed sufficiently stable dimers so that further release of p66*M* becomes extremely slow.

The dimerization process in this study also allows identification of additional intermediate resonances. Additional thumb′ resonances for Ile274′ and Ile257′ (**Figure 8—figure supplement 1**) exhibit an initial intensity increase and subsequent decrease, consistent with conformational intermediates. Although it is not clear if the two intermediate states are also present in the p66/p66′ homo-dimerization study, a close examination of the same region of the spectrum suggests that similar intermediate species may be present. Given the involvement of the thumb′ in both early and late conformational events, this behavior is probably not surprising.

In summary, the dimerization of [$^{13}$CH$_3$-Ile] p66ΔPL with p66 is qualitatively similar to p66 homodimerization, but the monomer is significantly more persistent and the time constants are all longer. The Ile47′ resonance provides a direct readout of dimer formation that probably is not limited by additional conformational changes.

## Maturation of helix αM′

The residues that form helix αM′ are almost all hydrophobic; the lone exception is Lys424, which also can interact hydrophobically with its (CH$_2$)$_4$ sidechain. This uniformity allows it to adopt alternate registrations in which one hydrophobic residue substitutes for another. This conformational variability is supported by a comparison of multiple crystal structures (**Figure 1—figure supplement 3**). The ability of the helix to adopt alternate registrations facilitates its victory in the tug-of-war for residues from RH′. Thus, immature, distorted helical conformations can be present that are more consistent with a folded RH′ domain, and the helix is then able to recruit and incorporate Tyr427′ from RH′ when this residue is released from RH′ due to thermal fluctuations. Recruitment of Tyr427′ into αM′ results in improved helical geometry and more stable interactions between αM′ and other connection′ residues. As shown previously, RH′ is significantly destabilized by the loss of Tyr427′, facilitating its unfolding and subsequent proteolytic degradation (**Zheng et al., 2014**).

## The conformational maturation process

The studies presented above support a modified conformational selection process and provide a basis for characterizing some of the steps in **Equation 1c** (**Figure 9**). The structure of the p66 monomer provides perhaps the most compelling support for a conformational selection model, since most of the domain interfaces are abrogated in the monomer without the need for dimer formation to promote this process. Only the fingers/palm:connection needs to dissociate to allow the necessary reorganization of the domains. The inherent preference of the bent fingers/palm domains to adopt a more extended conformation provides some additional impetus for dissociation of this interface (**Figure 3**). Further support for this model is derived from the effect of the palm loop deletion in blocking dimerization and the molecular dynamics simulations presented above (**Figure 4**). The initially formed homodimer contains two folded RH domains and two immature connection domains. The fingers/palm domains in the two subunits are probably close to their final conformation in the initial homodimer, since the initial isomerization of the monomer to the extended conformation is proposed to be concerted with straightening of the fingers/palm (**Figure 4**). The RH:thumb′ interface

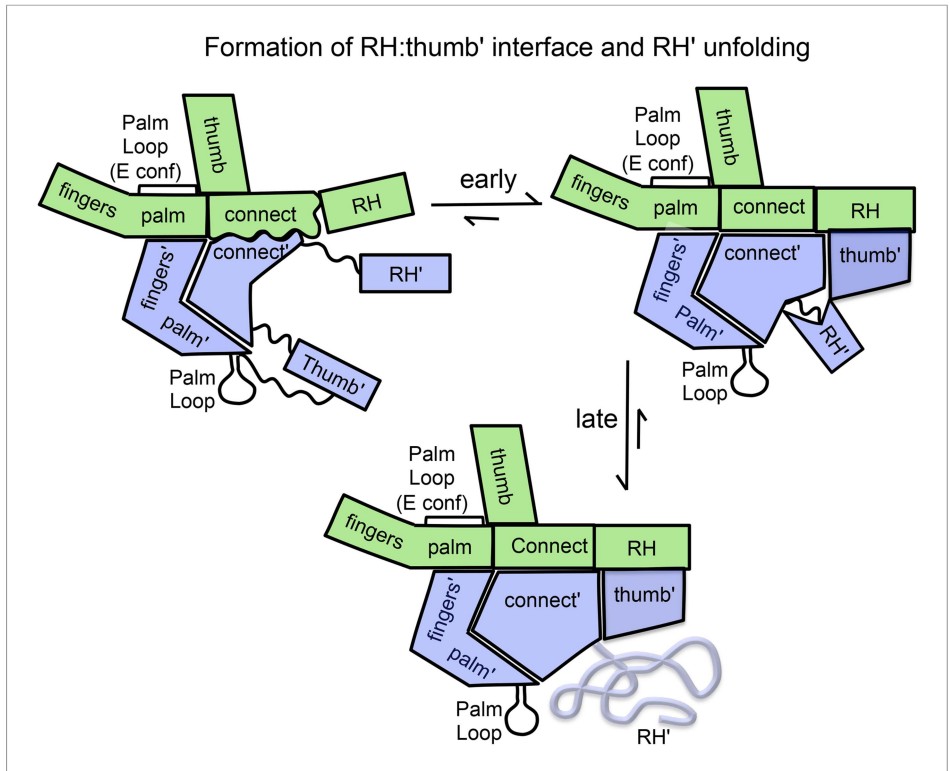

**Figure 9**. Schematic illustration of the maturation of the p66/p66′ homodimer. This figure illustrates the more rapid and the slower time-dependent changes occurring subsequent to initial isomerization/dimerization. The subunit conformations are color coded as in *Figure 1*.

is not initially present. Based on the analysis of Ile methyl resonances of residues distributed throughout the molecule, slower processes that nevertheless are largely accomplished during the first HMQC accumulation period include: 1) formation of the inter-subunit RH:thumb′ interface; 2) maturation of the connection:RH interface; 3) maturation of the p66 connection and thumb subdomains. These conformational steps appear to be cooperative.

Substantial NMR evidence indicates that several other conformational processes occur on a much slower time scale. These include: 1) transfer of residues from RH′ to the connection′ domains of p66′, 2) conformational maturation of the p66′ connection′ and thumb′ domains, including formation and/or extension of helix αM′, and 3) unfolding of the destabilized RH′ domain. Resonance changes related to these slower processes are readily observed in successive NMR spectra (*Figures 7 and 8* and *Tables 1 and 2*) and result from the gradual transfer of residues from RH′ to helix αM′ in the connection′ domain, which affect primarily the resonances of residues located in the RH′, thumb′, and connection′ domains of the p66′ subunit.

The time constants describing RH′ unfolding are similar to the 7-hr lifetime estimated for the HIV virion based on mathematical models (*Perelson et al., 1996*; *Perelson and Nelson, 1999*). Additional stabilization of the initial p66/p66′ dimer species likely results from complex formation with tRNALys,3-annealed viral genome, which is present in the virion core (*Kleiman et al., 2010*). Formation of such a complex is not expected to impact the maturation pathway described above, but to accelerate the process by eliminating dissociation of immature dimers and introducing additional stabilizing interactions for the E conformer. Preliminary NMR studies demonstrate that addition of dsDNA can promote dimerization and facilitate maturation.

In summary, the metamorphic polymerase domain of RT can be considered as a puzzle with two alternate solutions. The monomer structure corresponds to a partially disassembled version of the puzzle, with only the fingers/palm:connection interface remaining, and is thus primed to undergo a unimolecular reorganization into either the compact or extended forms, facilitating dimer formation and followed by conformational maturation.

# Materials and methods

## Protein expression and purification

All protein expression and purification followed the protocols described in our previous study (*Zheng et al., 2014*). All mutations used for resonance assignments of connection and RH domain were carried out by the QuickChange XL site-directed mutagenesis kit (Stratagene) and confirmed by DNA sequence analysis. Labeled proteins were prepared by growth on M9 minimal medium in 99% $D_2O$ supplemented with 50 mg/L [4-$^{13}$C,3,3-$^2H_2$]2-oxobutyrate 1 hr prior to induction as described previously (*Tugarinov and Kay, 2003*; *Zheng et al., 2014*). The [U-$^2$H,δ-$^{13}CH_3$-Ile] labeling pattern that is produced using this approach is abbreviated as [$^{13}CH_3$-Ile] throughout the manuscript and in the Supplementary figures. Mutants used for site-specific assignments are summarized in *Figure 5—figure supplement 4*.

## NMR spectroscopy

The $^1$H-$^{13}$C HMQC spectra were obtained using Agilent's gChmqc experiment in Biopack (Agilent, Santa Clara, CA). The NMR data were collected on a UNITY INOVA 800 MHz spectrometer equipped with a 5-mm Varian $^1$H[$^{13}$C,$^{15}$N] triple-resonance cryogenically cooled probe at 25°C or 35°C. In the $^1$H dimension, 1024 complex points were acquired with a sweep width of 14 ppm using a relaxation delay of 2 s. In the indirect $^{13}$C dimension, 96 complex points were acquired with a spectral width of 10 ppm, and the $^{13}$C offset was set to 13 ppm. A WURST-80 decoupling sequence was used for $^{13}$C-decoupling during the acquisition period (*Kupce and Freeman, 1995*). The residual water peak was suppressed using the WET sequence at the end of the relaxation delay (*Smallcombe et al., 1995*). All NMR data were processed by NMRPipe (*Delaglio et al., 1995*) and analyzed with NMRViewJ (*Johnson and Blevins, 1994*). The NMR samples were concentrated to 270 µL using Amicon ultracentrifugal filters with a 30 kDa cutoff, into the $D_2O$ NMR buffer: 25 mM Tris-HCl-d11, pD 7.5, 50–100 mM KCl, and 10–30 µM DSS as a chemical shift reference.

## Gel filtration analysis

The purified p66 and p66ΔPL were analyzed on the HiLoad 26/60 superdex 200 column separately. The running buffer was 50 mM Tris–HCl, pH 8.0, 200 mM NaCl, 1 mM ethylenediaminetetraacetic acid (EDTA) at a flow rate of 0.5 ml/min on an Akta FPLC system at 4 °C. The elution profiles recorded the absorbance at 280 nm.

## Dimerization studies

For the time-dependent NMR studies of the dimerization of unlabeled p66 with [$^{13}CH_3$-Ile]p66ΔPL, we mixed the labeled p66ΔPL with a twofold excess of unlabeled p66, concentrated the sample, and exchanged it into the NMR buffer: 25 mM Tris-HCl-d11, pD 7.5, 100 mM KCl, and 0.02% $NaN_3$, with Amicon Ultra Centrifual Filters (30 Kda cut-off). The final 275 µL sample contained 45 µM [$^{13}CH_3$-Ile] p66ΔPL and 90 µM of unlabeled p66. Successive $^1$H-$^{13}$C HMQC spectra were obtained in 5.5-hr increments, as described in our previous study (*Zheng et al., 2014*).

To prepare the labeled p51/p51' homodimer, we concentrated [$^{13}CH_3$-Ile]p51 and exchanged it into 25 mM Tris-HCl-d11 in $D_2O$ (pD = 7.5), 800 mM KCl, 20 mM $MgCl_2$, and 0.02% $NaN_3$ to a final concentration of 150 µM [$^{13}CH_3$-Ile]p51. It was necessary to use the higher salt conditions to compensate for the weak homodimerization constant of p51.(*Venezia et al., 2006*; *Marko et al., 2013*) The $^1$H-$^{13}$C HMQC spectra indicate that the sample is ∼90% dimer/10% monomer.

For the studies of the mutated heterodimer, [$^{13}CH_3$-Ile]p66/p51(L289K), we mixed a twofold excess of unlabeled p51(L289K) with [$^{13}CH_3$-Ile]p66 and exchanged the sample into 25 mM Tris–HCl in $D_2O$ (pD = 7.5), 50 mM KCl, and 0.02% $NaN_3$ to get 45 µM [$^{13}CH_3$-Ile]p66/p51(L289K) samples.

## Assignments of connection and RH domain resonances

In our previous study (*Zheng et al., 2014*), we utilized constructs of the isolated fingers/palm, RH, and thumb domain to assign many of the isoleucine δ-$CH_3$ resonances in RT. Several preliminary connection domain assignments were also derived from site-directed mutants. In the present study, we report more complete assignments for the connection and RH domain resonances based on extensive mutagenesis studies (*Figure 5—figure supplements 3–15*). In a few cases, these resulted in

assignment changes. The analysis presented previously was not dependent on these assignments, and the earlier conclusions are unaffected by the reassignments.

## Molecular dynamics simulations

Molecular dynamics simulations were performed on the isolated fingers/palm domain, defined to include residues 1–236, starting with either subunit of the RT heterodimer, pdb: 1DLO (*Hsiou et al., 1996*). Since the segment from 219–230 is missing in the p51 subunit of 1DLO, the missing residues were modeled by using the corresponding segment from the p66 subunit. The ends of the palm loop are separated by 20 Å in p66 compared with 7.2 Å in p51, so that this insertion leads to a localized perturbation. However, the initially increased separation of residues 218 and 231 required for the segment transplant decayed during the first 10 ns equilibration period, and the time-dependent simulations shown in the figures begins at the end of this period.

The structures were solvated in a box of water (p51 with 24,721 and p66 with 26,635 water molecules, respectively), after missing protons were added to each of these structures. Prior to equilibration, all systems were subjected to (i) 100-ps belly dynamics runs with fixed peptide, (ii) minimization, (iii) low-temperature constant pressure dynamics at fixed protein to assure a reasonable starting density, (iv) minimization, (v) stepwise heating molecular dynamics at constant volume, and (vi) constant volume molecular dynamics for 5 ns. All final unconstrained trajectories were calculated at 300 K under constant volume (100 ns, time step 1 fs) using the PMEMD module of Amber (*Case et al., 2005*) to accommodate long-range interactions. The parameters were taken from the FF10 force field of Amber (*Case et al., 2005*). An additional 300-ns trajectory for the p51 system was calculated with a different set of starting velocities.

Similar calculations were also performed starting with a p51 subunit of a structure that included the palm loop residues, pdb: 1S9E (*Das et al., 2004*), and starting with the monomer, pdb: 4KSE (*Zheng et al., 2014*). Since in the monomer construct residues 218 and 231 are directly bonded, two alternate procedures were used to introduce the missing palm loop residues. Either the segment from the p66 subunit was introduced, analogous to the procedure described above, or the artificial bond was left in place and 13 additional residues were included to maintain the same total number of residues (237). The results of these simulations are shown in *Figure 3—figure supplement 1*.

## Quantitative data analysis

Time-dependent intensity data obtained in studies of p66/p66′ and p66/p66ΔPL were analyzed using the non-linear least squares feature of Mathematica (Wolfram Research). Time-dependent intensity data were fitted to growing or decaying exponential functions that also allowed for variable limiting values for data sets that could not be well approximated by a transition between fractional probabilities of 0 and 1 (*Tables 1 and 2*). For Ile47, the t = 0 intercepts were normalized to total 1.0, and the fits demonstrated that the Ile47M + Ile47C summed intensity was nearly constant. Although for some of the resonances analyzed in the p66/p66ΔPL dimerization study, there is significant overlap between the monomer and dimer species, no correction for this overlap was utilized.

## Acknowledgements

FUNDING: National Institute of Health (Intramural Research Program); National Institute of Environmental Health Sciences (Research Project Number Z01-ES050147 to REL); National Institutes of Health, NIEHS (Delivery Order HHSN273200700046U to EFD). The authors are grateful to Drs Juno Krahn, Jason Williams, and Lars Pedersen for many helpful discussions.

## Additional information

### Funding

| Funder | Grant reference | Author |
|---|---|---|
| National Institute of Environmental Health Sciences (NIEHS) | Z01-ES050147 | Robert E London |

| Funder | Grant reference | Author |
|--------|-----------------|--------|
| National Institute of Environmental Health Sciences (NIEHS) | HHSN273200700046U | Eugene F DeRose |

The funders had no role in study design, data collection and interpretation, or the decision to submit the work for publication.

## Author contributions

XZ, Conception and design, Acquisition of data, Analysis and interpretation of data; LP, Conception and design, Analysis and interpretation of data; GAM, EFD, REL, Conception and design, Analysis and interpretation of data, Drafting or revising the article

# Additional files

## Major datasets

The following previously published datasets were used:

| Author(s) | Year | Dataset title | Dataset ID and/or URL | Database, license, and accessibility information |
|-----------|------|---------------|------------------------|--------------------------------------------------|
| Hsiou Y, Ding J, Das K, Clark Jr. AD, Hughes SH, Arnold E | 1996 | Human immunodeficiency virus type 1 | http://www.rcsb.org/pdb/explore/explore.do?structureId=1DLO | Publicly available at RCSB Protein Data Bank (1DLO). |
| Das K, Clark AD, Lewi PJ, Heeres J, De Jonge MR, Koymans LM, Vinkers HM, Daeyaert F, Ludovici DW, Kukla MJ, De Corte B, Kavash RW, Ho CY, Ye H, Lichtenstein MA, Andries K, Pauwels R, Boyer PL, Clark P, Hughes SH, Janssen PA, Arnold E | 2004 | Crystal structure of HIV-1 reverse transcriptase (RT) in complex with Janssen-R129385 | http://www.rcsb.org/pdb/explore/explore.do?structureId=1S9E | Publicly available at RCSB Protein Data Bank (1S9E). |
| Zheng X, Pedersen LC, Gabel SA, Mueller GA, Cuneo MJ, DeRose EF, Krahn JM, London RE | 2014 | Crystal structure of a HIV p51 (219-230) deletion mutant | http://www.rcsb.org/pdb/explore/explore.do?structureId=4KSE | Publicly available at RCSB Protein Data Bank (4KSE). |
| Unge T, Knight S, Bhikhabhai R, Lovgren S, Dauter Z, Wilson K, Strandberg B | 1994 | 2.2 angstroms resolution structure of the amino-terminal half of HIV-1 reverse transcriptase (fingers and palm subdomains) | http://www.rcsb.org/pdb/explore/explore.do?structureId=1HAR | Publicly available at RCSB Protein Data Bank (1HAR). |

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
