## [Decision Letter]

Thank you for sending your work entitled "Asymmetric conformational maturation of HIV-1 reverse transcriptase" for consideration at *eLife*. Your article has been favorably evaluated by John Kuriyan (Senior editor) and three reviewers, one of whom, Volker Dötsch, is a member of our Board of Reviewing Editors.

The Reviewing editor and the other reviewers discussed their comments before we reached this decision, and the Reviewing editor has assembled the following comments to help you prepare a revised submission.

Zheng et al. describe a detailed investigation of the conformational changes of the reverse transcriptase of the HIV virus. This protein forms dimers, which however, change their conformation to become asymmetric. The authors use MD simulation to obtain insight into the fast events of this conformational changes as well as NMR spectroscopy to follow the slower events. Due to the large size of the proteins investigated only methly group labeling is possible. Overall this is a very interesting study that provides insight into a very important process. The conformational changes in the HIV reverse transcriptase are important from a medical as well as from a basic science view.

The main criticism of all three reviewers is that the paper does read somewhat densely, especially for somebody who is neither an NMR expert nor familiar with the architecture of RT and it is suggested that the manuscript is significantly reduced in size. The results from [33] form the basis for most of the Results and Discussion, and thus are extensively discussed throughout the manuscript. For the acceptance in *eLife* it will therefore be important to clearly focus on the new data and clearly indicate what is new relative to the previous published manuscripts. Summaries and discussion of previously published data should be shortened.

Additional major points:

1) The readability of the paper would be improved by including in each figure presenting an NMR spectrum a schematic of what molecular complexes are being compared and where the labeling is. This will help the reader understand the experiments, especially since there are a lot of different protein states discussed in the paper.

2) For a direct comparison it would also be helpful to provide in addition to the overlay of the spectra the individual ones to better be able to judge peak overlap and spectral quality (sup mat).

3) The spectra presented in Figures 5, 6, 7 and 8 are intriguing and the analyses are relevant. However, the p66/p66' spectra for the RH region based on which the authors propose RH-RH' interaction is not highly convincing. The authors may elaborate on this part, particularly the basis for RH-RH' interaction.

4) Based on the current and earlier studies by the authors, the region 218-231 appears to be critical for E conformation. In the light of the MD study the authors need to discuss the conformations of the region in p66 and p51 and their possible implications for folding.

5) The fits of the kinetic data shown in the supplementary material section show a substantial scattering of the data making it unclear how accurate the data are. The error for the kinetic constants should be provided and discussed.

6) Does the kinetics of the observed maturation fit to the lifetime of the protein in infected cells? The lifetime should be significantly longer than the time it takes for the protein to adopt the active conformation.

---

## [Author Response]

*1) The readability of the paper would be improved by including in each figure presenting an NMR spectrum a schematic of what molecular complexes are being compared and where the labeling is. This will help the reader understand the experiments, especially since there are a lot of different protein states discussed in the paper*.

I believe that in the revised version of this manuscript we have done a much better job of clearly defining what is contributed by this study and we have eliminated a substantial amount of material reviewing the results covered in our previous study. This study is focused on a determination of the RT maturation process that converts the initial p66 monomer into the p66/p51 heterodimer, employing NMR and molecular dynamics to provide insights into the individual steps involved. Importantly, since all current RT inhibitors target the mature structure, understanding the maturation process, which can involve more fragile structural intermediates, opens the possibility of interfering with maturation rather than with function, which in principle also eliminates the problem of trying to compete with the substrates for binding. This is a rather complex molecule to evaluate and to describe, but the improved readability was apparent to all who read this revision.

*2) For a direct comparison it would also be helpful to provide in addition to the overlay of the spectra the individual ones to better be able to judge peak overlap and spectral quality (sup mat)*.

The suggestion to include additional schematics was excellent, and indeed, it became apparent to us as we went through the revision that this provides a major clarification of the presentation.

*3) The spectra presented in*
Figures 5, 6, 7 and 8
*are intriguing and the analyses are relevant. However, the p66/p66' spectra for the RH region based on which the authors propose RH-RH' interaction is not highly convincing. The authors may elaborate on this part, particularly the basis for RH-RH' interaction*.

The major spectral overlay problem occurs in Figure 5, and we have included two additional supplementary figures (Figure 5—figure supplement 1 and Figure 5—figure supplement 2) that show the spectra separately. Again, we agree that this is a very useful clarification.

The reviewers’ comment regarding our conclusions about the RH:RH' interaction was the only one that we had some difficulty with. Despite multiple readings by all of the coauthors, we were unable to understand the nature of the request. As far as we are aware, there is no direct interaction between the two RH domains, and we did not mean to imply otherwise. We have tried to revise the presentation to further clarify the interactions for which we do find evidence. The intersubunit interaction between the thumb' and the RH domain is, we believe, significant, and is supported most directly by the study utilizing the L289K' mutation in the p51 subunit of RT.

*4) Based on the current and earlier studies by the authors, the region 218-231 appears to be critical for E conformation. In the light of the MD study the authors need to discuss the conformations of the region in p66 and p51 and their possible implications for folding*.

The reviewers’ comment asking about the behavior of the palm loop residues in the simulations was extremely useful, causing us to more carefully evaluate an aspect of the molecular dynamics simulations that we had overlooked. A detailed analysis revealed that although all simulations showed the increased A/F angle (Figure 3 and Figure 3—figure supplement 1), the limiting conformations do not uniformly approach the conformation that is observed in the p66 subunit of RT. The approach utilizing truncation of the modeled sequence at residue 236 in order to reveal the intrinsic conformational preferences of the fingers/palm eliminates direct interactions with other domains that are important for determining the fingers/palm conformations at the domain boundaries. Further, we found that although the distance between residues Asp218 and Gly231 that flank the palm loop does increase during the course of the simulations, this change is not associated with formation of the mature primer grip conformation that is present in the p66 subunit of RT. In addition to the absence of inter-domain interactions, termination of the sequence at residue 236, allows the C-terminal residues to adopt a more unconstrained ensemble of positions that are not representative of the full RT molecule. To summarize, we believe that truncation of the sequence after residue 236 is a useful approach for evaluating the intrinsic domain orientation of the fingers/palm. However, the deletion eliminates interactions with other domains that can significantly affect the conformation of residues near the domain boundaries, leading to some artifactual results. In the revised manuscript, in the last paragraph of the subsection entitled “Intrinsic conformational preferences of the fingers/palm”, we have dropped the distance calculation from the paper and added some discussion to clarify what features of the fingers/palm are and are not explained by the simulations.

*5) The fits of the kinetic data shown in the supplementary material section show a substantial scattering of the data making it unclear how accurate the data are. The error for the kinetic constants should be provided and discussed*.

In the revised version of the manuscript, mean values corresponding to apparent kinetic time constants are summarized in Tables 1 and 2, and the standard error determination is based on at least three measured growth or decay curves. Prompted in part by the reviewers’ questions, we made an extensive review of the kinetic analysis and for the data in Table 2, decided to adopt a more phenomenological approach in which we selected the simplest model that most closely approximated the data and did not try to introduce corrections for resonances in which there is significant monomer/dimer overlap. This decision was based on the fact that in the dimerization study with p66∆PL, the there is a relatively large fraction of monomer species, making the correction as large as the measured data, as well as the fact that in subtracting different resonances, it becomes necessary to introduce an arbitrary correction factor to account for differences in relaxation behavior. The revised analysis for Ile47 is particularly informative leading to the conclusion that a more rapid initial dimerization accounting for about 20-30% of the monomer is followed by a slower dimerization reaction with a time constant similar to that of most of the slow time constants that we measure. We also have included multiple examples of the kinetic fits as supplementary figures to Figure 7 and Figure 8 to illustrate the quality of the data fits. One other change is that we dropped one of the four data sets used in the homodimerization study, so utilized three rather than the original four studies. In the study that was dropped, the labeling level was significantly decreased due to an error in addition of the labeled precursor, so that the S/N was generally much lower.

*6) Does the kinetics of the observed maturation fit to the lifetime of the protein in infected cells? The lifetime should be significantly longer than the time it takes for the protein to adopt the active conformation*.

The best estimates of the virion lifetime are from the analyses published by Perelson and colleagues, which are now referenced in the paper. He reports mean virion lifetimes of ∼ 7 h, which is a bit longer than the slower time constants that we have obtained in the homodimerization study. According to [16], the viral core contains a complex formed from tRNALys, 3-annealed viral genome plus reverse transcriptase (see Figure 1 of this reference). Formation of this complex will help to stabilize the dimer form and can facilitate maturation. Preliminary NMR studies demonstrate that addition of dsDNA exerts this type of effect. Importantly, formation of this type of complex should not alter the nature of the maturation steps that we have observed, but only help them to occur on a shorter time scale.